# Knowledge, Attitude, and Practice towards COVID-19 among Patients Attending Phuentsholing Hospital, Bhutan: A Cross-Sectional Study

**DOI:** 10.3390/ijerph20042942

**Published:** 2023-02-08

**Authors:** Kinley Gyeltshen, Sangay Phuntsho, Kinley Wangdi

**Affiliations:** 1Gidakom Hospital, Gidakom, Thimphu 11006, Bhutan; 2Phuentsholing General Hospital, Phuentsholing 21101, Bhutan; 3Department of Global Health, National Centre for Epidemiology and Population Health, College of Health and Medicine, Australian National University, Acton, Canberra, ACT 2602, Australia

**Keywords:** Bhutan, Phuentsholing, COVID-19, knowledge, practice, attitude

## Abstract

Bhutan is one of the few countries in the world to take unprecedented steps to control the spread of COVID-19 in the country. This study aimed to investigate knowledge, attitude, and practice (KAP) and their associated covariates among patients attending Phuentsholing Hospital, Bhutan. Therefore, a cross-sectional study was conducted among patients attending Phuentsholing Hospital in Bhutan between March 17 and April 9, 2021, using an interview-administered questionnaire. The multivariable logistic regression was used to identify statistically significant covariates of good KAP. Further, the association between levels of KAP scores was assessed using Pearson’s correlation coefficient. Of the 441 participants, 54.6% (241) were female. Knowledge, attitude, and practice score were reported by 55.3%, 51.8%, and 83.7% of participants, respectively. Higher education, secondary education, monastic education, and non-formal education were 9 [adjusted odds ratio (AOR) = 9.23; 95% confidence interval (CI) 3.438, 24.797], 3.5 (AOR = 3.5; 95% CI 1.425, 8.619), and 4 (AOR = 3.8; 95% CI 1.199, 12.141) times more likely to report good knowledge than illiterates. A positive attitude was associated with higher (AOR = 2.97; 95% CI 1.154, 7.66) and secondary (AOR = 3.53; 95% CI 1.454, 8.55) education compared to illiteracy. The good practice was associated with higher (AOR = 12.31; 95% CI 2.952, 51.318) and secondary (AOR = 11.5; 95% CI 3.439, 38.476) education compared to illiteracy. Participants in the age groups 26–35 years (AOR = 0.11; 95% CI 0.026, 0.484) and >45 years (AOR = 0.12; 95% CI 0.026, 0.588) were less likely to exhibit good practice compared to those aged 18–25 years. Those working in the private or business sectors were 9 (AOR = 8.81; 95% CI 1.165, 41.455) times more likely to have good practice compared to civil servants. There was a weak but positive correlation between knowledge-attitude (r = 0.228), knowledge-practice (r = 0.220), and attitude-practice scores (r = 0.338). The need for health education on COVID-19 to increase knowledge and attitude is highly recommended, and should be focused on the less educated and other vulnerable groups such as farmers and students, as well as those older than 25 years.

## 1. Introduction

The Coronavirus 2019 (COVID-19), caused by the new coronavirus strain SARS-CoV-2, has become a serious public health problem globally [1,2,3]. The total cases have crossed 651.9 million, with 6.6 million deaths as of 23 December 2022 [3]. The pandemic has burdened all sectors, particularly health, causing infrastructural and manpower shortages and burnout in hospitals across the world [4,5]. In struggling to achieve universal health coverage, low- and middle-income countries are overburdened by the unprecedented challenge for health care systems and social policies [6]. The Omicron newer strains (e.g., Omicron sub-lineages) are emerging despite high vaccine coverage [7,8,9]. Therefore, compliance and adherence to non-pharmacological measures, such as hand hygiene, face mask use, and social distancing to prevent contracting the SARS-CoV-2 infection, will continue for some time to come. Compliance with the non-pharmacological control measures is significantly influenced by people’s knowledge, attitude, and practices (KAP) towards diseases [10,11,12]. In addition, a higher level of knowledge correlated with a positive attitude and good practice toward COVID-19 [13,14,15,16,17,18].

As of December 27, 2022, Bhutan had reported 62,524 COVID-19 cases with 21 deaths, and 2 million vaccine doses had been administered to date [3]. Further, the government has continued to advocate health measures including hand washing, social distancing, and the use of face masks in public places during the initial phase of the pandemic [14,15]. In addition, several facilities for hand washing were set up in schools and institutes across the country [16,17]. With the reports of the BF.7 variant of the Omicron in India in recent days, Bhutan reinstated the mandatory wearing of face masks in public areas and strongly encouraged hand hygiene [19]. Bhutan was one of the few countries in the world to take unprecedented steps to control the spread of COVID-19 in the country. All returning travelers were required to undergo a mandatory 21-day quarantine [20,21], which was longer than the 14 days practiced in other countries around the world. There have been two national lockdowns and several local/regional lockdowns in response to community transmission. The towns and districts along the Indian border have been identified as “red zones,” and a 7-day quarantine was required for people traveling out of these red zone areas [22].

Bhutan has successfully controlled the COVID-19 pandemic through high coverage of vaccination (>96% of the population vaccinated) and effective implementation of non-pharmacological control measures. However, no studies have been undertaken to understand the compliance and adherence of non-pharmacological control methods. In addition, a number of factors outside the health system, including socio-economic conditions, demographic patterns, family patterns, and the cultural and social fabric of societies’ sociopolitical and economic changes, determine the uptake of health education and behavior changes [11,23,24]. Therefore, it is imperative to undertake studies to understand the uptake of health education for adapting health education specific to local context. Further, to address the aforementioned paucity of information, the aims of this study were to assess KAP about COVID-19 among the patients attending Phuentsholing Hospital, Bhutan, and identify associated covariates for good KAP levels.

## 2. Methods

### 2.1. Study Design and Participants

A cross-sectional study was conducted among patients attending Phuentsholing hospital in Phuentsholing municipality under Chhukha district between 17 March and 9 April 2021 (Figure 1). The study population included both the residents as well as the migrant population of Phuentsholing municipality, Bhutan. Phuentsholing municipality is the commercial hub of Bhutan and shares a border with the Indian town of Jaigaon in West Bengal State, India. In 2017, the population of Phuentsholing was estimated to be around 25,000 [25]. Phuentsholing hospital caters to the neighboring sub-districts in Chhukha district and serves as a referral center for five primary health units (PHUs) in addition to the population of the municipality.

### 2.2. Sampling and Participant Recruitment

A sample size of 441 participants was calculated with a 50% probability of having good KAP towards COVID-19 using the following formula.
$$n=\frac{Z^{2}\;\cdot \;p\left(1-p\right)\;}{d^{2}\;}$$
where

*n* = required sample size,

*Z* = 1.96, taking a 95% confidence interval from the Normal table, two-tailed

*p* = expected proportion, 0.5

*d* = absolute precision, 0.05

Allowing for a 15% dropout rate, the sample size is 441.

The study participants were recruited using a simple random sampling approach. Every weekday, 20 outpatient department (OPD) registration numbers were randomly selected. The patients with these OPD numbers were invited to participate in this study. If the randomly selected patients refused to be interviewed, the patient with the next OPD registration number was invited for the study. Inclusion criteria were: Bhutanese, both sexes, 18 years of age or older, and willing to be interviewed for the study. Exclusion criteria were: Non-Bhutanese, refusing to be interview; seriously ill patients; patients under the age of 18 years.

### 2.3. Data Collection Instruments

The study tool was an interviewer-administered questionnaire that had five parts. Part A included sources of information about COVID-19. Part B contained 26 questions on knowledge. We assessed knowledge in domains such as transmission, symptoms, and prevention of COVID-19. The correct responses were given a score of 1, as done by Olum et al. [26], and a total score of ≥60% was identified as good knowledge (1 = good knowledge and 0 = poor knowledge). Part C of the questionnaire has 15 questions on attitude. The attitude questions had 5 responses: “strongly agree, agree, neutral, disagree, and strongly disagree.” The 5-point Likert scale of +2 to −2 was used against each attitude question. The responses were assessed using a 5-point Likert scale, as done by Goni et al. [27]. The scoring system for the 13 questions on attitude was +2 to −2, and +1 to −1 for the remaining two questions; therefore, the total score for attitude was 28. A total score of ≥60% was identified as a positive attitude and the rest as a negative attitude (1 = positive attitude and 0 = negative attitude). Part D has nine practice questions on a 5-Likert scale with a maximum score of 34. Responses were awarded a score of 4 to 0 for always, usually, sometimes, rarely, and never/don’t know, respectively. A total score of ≥60% was identified as having a good practice score (1 = good practice and 0 = poor practice) [16,27]. Finally, Part E has questions related to socio-demographic characteristics, including age, sex, level of education, occupation, and approximate monthly income.

The questionnaire was developed based on a literature review of different studies that were relevant to our study aims [13,28]. The final questionnaire was reviewed by a team of physicians from the Jigme Dorji Wangchuck National Referral Hospital (JDWNRH), and the physicians felt the questionnaire captured all aspects of COVID-19.

### 2.4. Data Processing and Analysis

The continuous variables were expressed as mean ± standard deviation (SD), and categorical data were expressed as a percentage. All KAP scores were classified as having a binary outcome-good vs. poor knowledge, positive vs. negative attitude, and good vs. poor practice. The univariate and multivariable logistic regression models were built using backward elimination for KAP to identify statistically significant covariates. Any variable with a *p* value < 0.2 in the univariate analysis, along with the main variable of interest, was considered a candidate variable in the final model. All potential dependent variables were entered in the full model, and adjusted odds ratios (AOR) with 95% confidence intervals (CI) were used to determine the correlates of each independent variable. All explanatory variables in the multivariable model were tested for multicollinearity using a variance inflation factor (VIF). The VIF < 10 was considered a good fit for regression analysis. The correlation between patient knowledge, attitude, and practice was investigated using Pearson’s correlation coefficient. A value of *p* ≤ 0.05 was considered statistically significant for both the multivariable model and the correlation analysis.

The questionnaire was coded and entered into Epi-data Entry version 3.1 (EpiData Association, Odense, Denmark) by two independent investigators (KG and SP), and errors were corrected by revisiting the original data and analyzing it in STATA Version 13 (Stata Corporation, Stata Statistical Software, licensed by Khesar Gyalpo University of Medical Sciences of Bhutan). The study map was created with ArcMap 10.5.1 (ESRI, Redlands, CA, USA).

## 3. Results

### 3.1. Socio-Demographic Characteristics of Study Participants and Source of Information on COVID-19

Out of a total of 441 samples interviewed, 54.6% (*n =* 241) were female. The mean age was 35 years, with an age range of 18–82 years. Lhotshampa ethnicity comprised 40.8% (180) of study participants, and 44.4% (196) had higher secondary education. The civil servants and (private/business/corporate) employees made up 56.5% (249) of the respondents (Table 1). The main source of COVID-19 information was television programs (70.8%, 312), followed by the internet and friends at 65.0% (286) and 40.8% (180), respectively (Figure 2). Bhutan Broadcasting Service, a state-owned television program, and the internet (the Prime Minister’s Office of Bhutan and the Ministry of Health Facebook page) were the most widely followed and reliable sources of information during the pandemic.

### 3.2. Knowledge

More than half of the respondents (55.3% or 244) fulfilled our set criteria of “good level of knowledge” with a mean score of 14.7 (SD = 6.5). The mean correct responses on transmission, symptoms, and prevention of COVID-19 were 59.2% (261), 61.4% (270), and 67.1% (272), respectively. About 80% (336) of the study participants correctly identified coughing/sneezing as the mode of virus transmission. The correctly responded symptoms were fever (86.4%, 349), dry cough (70.5%, 285), and shortness of breath (76.7%, 310). The correct responses to prevention questions were washing hands (87.4%, 376), wearing a mask (83.0%, 357), and practicing physical distancing (76.7%, 330). Half of the participants had wrongly responded that COVID-19 can be treated, and 11.6% (51) stated that COVID-19 is not a serious disease. The details of the proportion of participants with correct responses to each knowledge item of the questionnaire are shown in Table 2.

### 3.3. Attitude

A mean attitude score of 16.0 (SD = 6.0) was reported with 51.0% (224) of respondents reporting a positive attitude. In addition, a total of 88.2% (389) were willing to get the COVID-19 vaccination in the future. Similarly, 85.0% (375) reported that they would be worried if infected with COVID-19. The majority (95.9%; 423) of participants had a positive attitude toward the use of face masks and physical distancing. Further, a positive attitude was noted toward handwashing, with almost (94.6%; 417) of respondents reporting that it was “very easy and easy” to wash hands. The reasons for these were: the easy accessibility and availability of handwashing stations in Phuentsholing (18.0%; 80); hand washing is our daily habit (14.5%; 64); and it is effortless and takes less time (9.3%; 41); however, one-third of the respondents (130) believed that it was safe to go out during the pandemic. A negative attitude was noted among 18.0% (79) of the respondents who disagreed or strongly disagreed with the existing rule of quarantine while traveling from the southern districts (the “Red Zone”) to other parts of the country (Figure 3).

### 3.4. Practice

The majority of the respondents (83.7%, 369) implemented good practices (≥20.4) with a mean score of 24.4 (SD = 4.9). Eighty-three percent (354) of respondents always used face masks while visiting a public place, and 73.7% (325) used them while being sick. Often, hand washing was practiced by 70.3% (310) and 24.3% (107) of respondents, respectively. Less than half (46.0%, 205) of participants used face masks at home. The Facebook pages of the Prime Minister’s Office and hospitals (57.8%, 255) and the Bhutan Ministry of Health (57.4%, 253) were the main sources of information for COVID-19. During the COVID-19 crisis, 57.8% (255) and 43.3% (253) of participants visited hospitals and flu clinics to seek help (Table 3).

### 3.5. Correlates of Good Knowledge, Positive Attitude, and Good Practice

In the multivariable logistic regression, participants with higher education, secondary education, and “monastic and non-formal education” (MNFE) were 9 (AOR = 9.23; 95% CI 3.438, 24.797), 3.5 (AOR = 3.5; 95% CI 1.425, 8.619), and 3.8 (AOR = 3.8; 95% CI 1.199, 12.141) times more likely to have good knowledge compared to the illiterates. Similarly, higher (AOR = 2.97; 95% CI 1.154, 7.66) and secondary (AOR = 3.53; 95% CI 1.454, 8.55) education were associated with a good attitude. The students were twice as likely to report a positive attitude (AOR = 2.48; 95% CI 1.010, 6.097) compared to civil servants. However, farmers were 66.0% (AOR = 0.34; 95% CI 0.132, 0.869) less likely to have a positive attitude than civil servants. Similarly, Sharchokpa ethnicity was 50.0% (AOR = 0.50; 95% CI 0.271, 0.934) less likely to report a positive attitude compared to Ngalong. The participants with higher (AOR = 12.31; 95% CI 2.952, 51.318) and secondary education (AOR = 11.5; 95% CI 3.439, 38.476) were 12 times more likely to engage in good practice compared to illiterate. Compared to civil servants, those working in the private or business sectors (AOR = 8.81; 95% CI 1.165, 41.455) were nine times more likely to report good practices. Participants in the age groups 18–25 years (AOR = 0.11; 95% CI 0.026, 0.484) and >45 years (AOR = 0.12; 95% CI 0.026, 0.588), other ethnicities (Mangdeps, Bumthamps) (AOR = 0.08, 95% CI 0.01, 0.646), and students (AOR = 0.17, 95% CI 0.031, 0.942), were less likely to engage in a good practice (Table 4). There was a weak but positive correlation between knowledge and attitude (r = 0.228, *p* < 0.001), knowledge and practice (r = 0.220, *p* < 0.001), and attitude and practice scores (r = 0.338; *p* < 0.001), respectively (Table 5).

## 4. Discussion

This is the first study on KAP’s association with COVID-19 in the general Bhutanese population. Good knowledge and a positive attitude were reported by around half of the participants, while 83% reported good practice. The factors associated with good knowledge were higher, secondary, monastic, and non-formal education. Significant covariates for positive attitude were higher and secondary education and students. However, farmers were less likely to report a positive attitude. Good practices were significantly associated with higher education, those working in private companies, and business personnel. However, age, ethnicity, and students were associated with poor practices. There was a weak but positive correlation between knowledge and attitude, knowledge and practice, and attitude and practice scores.

Despite widespread awareness campaigns by the Royal Government of Bhutan through all the available mass media, including newspapers, television, radio, and various social media (Facebook), only half of the respondents had good knowledge and a positive attitude. This is lower in comparison to other published papers in the region [13,28,29,30,31,32], while it is in accordance with other studies [17,33,34]. The low knowledge and attitude scores are concerning because they indicate a gap in knowledge and attitudes related to COVID-19. Despite high coverage of vaccination, practicing public health measures toward COVID-19 will continue until the pandemic ends [35]. Most participants had misconceptions about the transmission of COVID-19. These included dirty hands and touching dry surfaces, which will not transmit COVID-19. Similarly, touching the face was not important in the prevention of transmission. Therefore, efforts to educate the public about COVID-19 and these misconceptions through the most widely used source of information [TV in our study] should be undertaken in earnest. In addition, monitoring the content of information on other open online media also needs to be undertaken. However, misinformation was disseminated through unofficial online media in other parts of the world, which led to distrust [36,37,38] and poor updates of vaccinations [39,40], and subsequently, to vaccination inequalities [41].

In this study, education level was a positive predictor of good knowledge. Participants with diplomas/degrees/master′s degrees, or higher degrees were associated with good knowledge of COVID-19. This is similar to the findings of other published papers [16,42,43,44]. The educated are better informed because they have better access to information through the internet, social media (Facebook and Twitter), or other technologies such as YouTube. This shows a substantial educational gap, possibly stemming from access to information or the content of information provided to the public [45]. Thus, the Bhutan Ministry of Health should customize the educational materials and make them understandable to the general masses through easily available sources such as national television [BBS TV].

A high proportion of good practices in this study can be explained by the Bhutan government’s unprecedented implementation of public health measures to safeguard citizens from COVID-19. These public health measures included the mandatory wearing of facemasks in public places, maintaining a social distance of 1.5 m, free distribution, and the mandatory provision of alcohol-based sanitizers in public places [46,47]. In addition, compliance with face mask use and social distancing was strictly monitored by police and civilian volunteers [*Desung*] [48]. However, positive practices toward COVID-19 such as social distancing, hand hygiene, and face masking should be reinforced and maintained in Bhutan to interrupt the transmission of the virus, especially with the recent increase of cases in the region [49].

A substantial number of sociodemographic factors significantly affected good practices, including education and working in private organizations and businesses. This finding was in keeping with observations made in several other published studies [11,16,26,34,50]. The private/business sectors were more likely than civil servants to engage in good practices for COVID-19 prevention. The COVID-19 pandemic in Bhutan disrupted economic activities due to lockdowns and restrictions on the movement of goods and services in the country. As a result, private and business people were affected most by this pandemic, economically. Therefore, this group of people could have adopted positive practices in the prevention of COVID-19 with the expectation of returning economic activities to normalcy.

It was observed that significant negative covariates of good practices were those aged 26–35 years, >45 years, and students. Older people engaging in poor practice have been reported in another study [51]. The other studies also reported good practices for >40 years [13,26,44,52]. It would be worthwhile to study this finding to determine whether it pertains to the medium of information dissemination through social media, including Facebook, websites, and other online platforms. This is particularly relevant to Bhutan, where older generations are more likely to be illiterate compared to younger generations. Similar to our study, students were less likely to engage in good practices toward COVID-19 prevention compared to civil servants in Saudi Arabia [28]. One plausible reason could be waning interest in engaging in COVID-19 prevention practices due to low or no local transmission in the community. During this study, no local transmission was reported in the country. However, it would be worthwhile to investigate the reasons for poor COVID-19 prevention practices among students because similar findings were reported in another study in Bhutan [53].

A total of over half of the study participants reported a positive attitude, which is much lower compared to another study from Bhutan [48] but similar to the finding of a study from Saudi Arabia [54]. The difference could be due to the study population because our study participants were from the general population, whereas Dorji et al.’s study participants were university students. This seems to be supported by our study findings, which showed that the positive predictors of positive attitudes were those with higher and secondary education and students. The farmers were less likely to report a positive attitude toward COVID-19. This is because the negative attitudes among farmers could be partly explained by the preventative public health measures, including lockdowns, social distancing, and wearing face masks. Implementing these measures is difficult due to the nature of their work. These measures have led to a global shortage of food [55].

There was a positive correlation between knowledge and attitude, knowledge and practice, and attitude and practice. The positive knowledge and attitude [15,45], attitude and practice [28,45,55,56], and knowledge and practice [15,45,55] were reported in an earlier study. Good knowledge was strongly associated with positive preventive behaviors for other diseases [57]. Having good knowledge can help in the early identification of diseases and lead to better health-seeking behavior. Knowledge affects an individual’s behavior, and a higher knowledge level reinforces healthier behaviors [58], including social distancing, mass gathering, and shaking hands [28]. Bhutan’s government should make efforts to increase knowledge of COVID-19 because public health measures still have a role in the prevention and slowing of the transmission of COVID-19 in the country.

In summary, the efforts to educate the public on COVID-19 through the most widely used source of information (TV in our study) should be undertaken in earnest. In addition, the Ministry of Health should customize the educational materials and make them understandable to the general masses through easily available sources such as national television (BBS TV). Further, positive practices toward COVID-19 such as social distancing, hand hygiene, and face masking should be reinforced and maintained in Bhutan to interrupt the transmission of the virus. In the future, it would be worthwhile to investigate the reasons for poor COVID-19 prevention practices among students because schools have reopened across Bhutan and they can serve as the source of transmission.

### 4.1. Conclusions

Positive practice was reported by two-thirds of the participants in Phuentsholing hospital. However, good knowledge and a positive attitude were reported by only half of the study participants. Low educational status was significantly associated with poor KAP. In addition, increasing age and being a student were negatively associated with good practice. Therefore, health education programs aimed at improving COVID-19 KAP—particularly improving misconceptions on transmission and prevention in a language easily understandable by the general public—should be continued. BBS TV can continue to be the main source of information because of its reach. Some categories of the population identified in this study may benefit from specific health education programs to raise COVID-19 knowledge and attitudes.

### 4.2. Limitations and Strengths of the Study

The results of this study should be interpreted in light of some limitations. First, the cross-sectional study design does not allow for causal inferences and therefore cannot be established. Second, this study was conducted in Phuentsholing, which is one of the main centers of quarantine locations. Hence, the public might be better educated than the general public of Bhutan. However, the study participants were all the people visiting Phuentsholing municipality. Third, social desirability might have led to reporting attitudes and practices expected from the community and government. Fourth, due to the evolving nature of the COVID-19 pandemic, some findings might have changed now. Despite these limitations, this is the first study on COVID-19 KAP in the general population of Bhutan. Therefore, the study findings will be useful to inform policymakers and healthcare professionals regarding the development of future public health interventions, awareness-raising efforts, and health education programs.

### 4.3. Practical and Theoretical Implications of the Work

The infectious diseases related to viral mutations, as evidenced by the emergence of several variants of coronavirus during the past few years, suggest that there should be a rock-solid foundation of public health strategies to prevent or counteract future outbreaks at the national level. The knowledge, attitude, and practice of any infectious disease, particularly viral or bacterial, are almost similar with regard to prevention and control measures. Therefore, the recommendations provided in the conclusion of this paper could prove useful in the post-pandemic era for adopting public health measures such as health education specific to the local context. 

## Figures and Tables

**Figure 1 ijerph-20-02942-f001:**
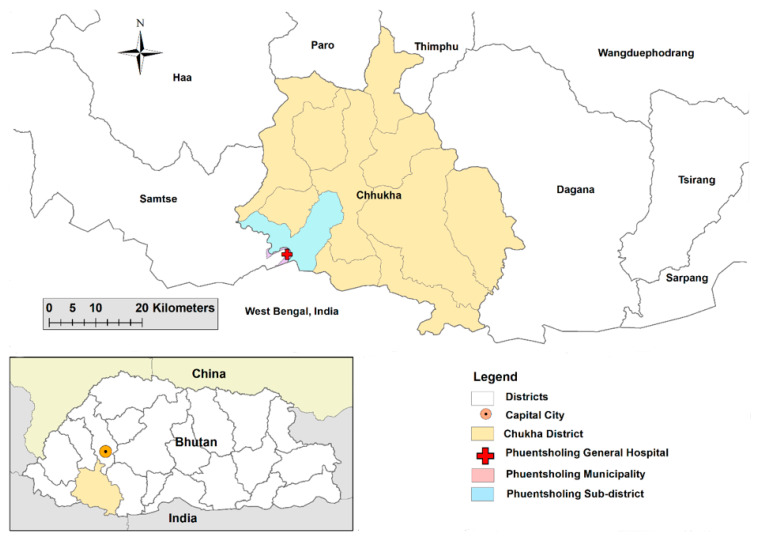
Map of Phuentsholing Hospital, Chhukha, Bhutan.

**Figure 2 ijerph-20-02942-f002:**
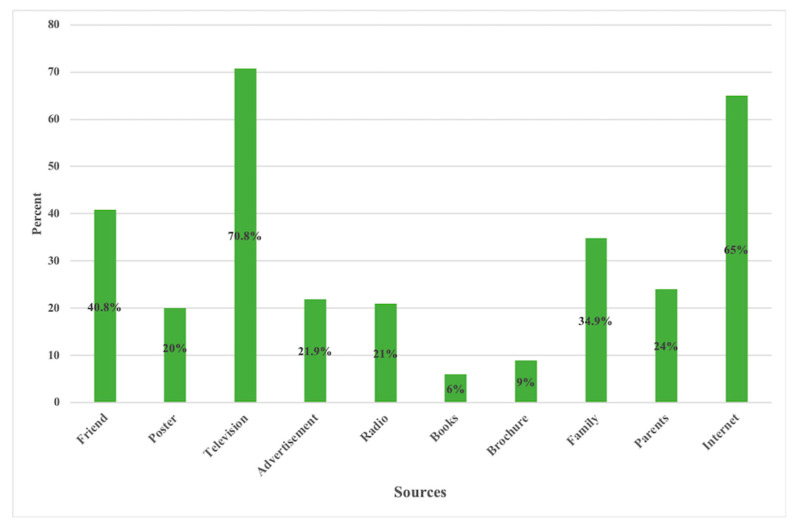
Different of sources of COVID-19 information among study participants in Phuentsholing Hospital, Bhutan.

**Figure 3 ijerph-20-02942-f003:**
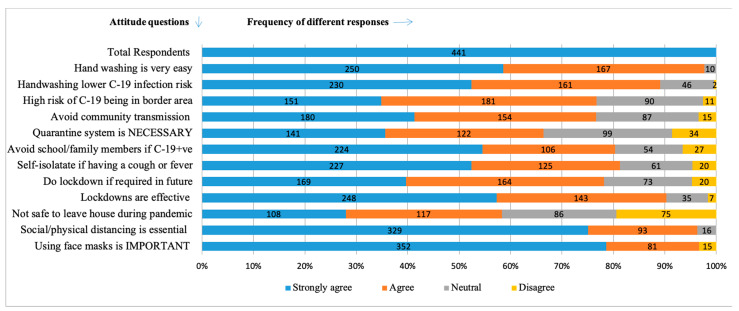
Frequency distribution of responses to attitude questionnaire among study participants in Phuentsholing Hospital, Bhutan (C-19 = COVID-19; +Ve = Positive).

**Table 1 ijerph-20-02942-t001:** Demographic characteristics of the participants in Phuentsholing Hospital, Bhutan.

Characteristics	Number	Percent
Sex		
Male	200	45.4
Female	241	54.6
Age groups(years)		
18–25	121	27.4
26–35	153	34.7
36–45	82	18.6
>45	85	19.3
Ethnicity		
Ngalong	80	18.1
Sharchokpa	146	33.2
Lhotshampa	180	40.8
Khengpa	24	5.4
Others ethnicity	11	2.5
Highest level of education		
Cannot read and write	68	15.4
MNFE *	26	5.9
Primary	22	5.0
Higher secondary	196	44.4
Higher education (Diploma, degree, or higher)	129	29.3
Occupation		
Civil servant	120	27.2
Private/corporate/business	129	29.3
Homemaker	44	9.9
Student	43	9.8
Farmer	68	15.4
Others **	37	8.4

* MNFE-monastic and non-formal education; ** Others-monks and armed forces.

**Table 2 ijerph-20-02942-t002:** Proportion of participants with good responses to the knowledge items of the questionnaire of study participants, Phuentsholing Hospital, Bhutan.

Question Theme (%) *	Responses	N (%)
Transmission (59.2%)	Dirty hands	196 (47.2)
	Touching dirty surfaces	125 (30.1)
	Spread via droplets from coughing of an infected person	336 (81.0)
Symptoms (61.4%)	Fever	349 (86.4)
	Dry cough	285 (70.5)
	Shortness of breath	310 (76.7)
	Feeling tired	197 (48.8)
	Sore throat	200 (49.5)
Prevention (67.1%)	Washing hands regularly	376 (87.4)
	Cleaning hands with hand sanitizer	339 (78.8)
	Wearing mask	357 (83.0)
	Not touching face	205 (47.5)
	Covering mouth when coughing	323 (75.1)
	Staying home	307 (71.4)
	Doing enough exercise	126 (29.3)
	Avoiding going to a crowded place	324 (75.4)
**Knowledge**		
Good knowledge	244 (55.3%)	
Mean ± SD	14.7 ± 6.5	
Range (min–max)	0–24	

N-number; * percent of respondents having a good score; SD-standard deviation.

**Table 3 ijerph-20-02942-t003:** Frequency and percentage of participants who opted for “always” or “usually” to the practice items of the questionnaire of the study participants, Phuentsholing Hospital, Bhutan.

Practice Questions		Always/ Usually (%)
P1. Do you cover your mouth when sneezing or coughing?		394 (89.8)
P2. Do you wash your hands regularly?		417 (94.6)
P3. Do you use soap while you wash your hands?		411 (93.2)
P4. Do you clean your hands with a hand sanitizer?		306 (69.4)
P5. While COVID-19 is still a problem in our city, do you wear a mask when you are sick?		383 (86.9)
P6. While COVID-19 is still a problem in our city, do people in your family wear a mask at home?		205 (46.5)
P7. During the COVID-19 crisis, do people in your family wear a mask when leaving home?		414 (93.9)
*Practice related to information and place of visit during COVID-19 Pandemic*		
P8. During the COVID-19 crisis, which websites do you usually rely on for correct information?	Prime Minister’s Office (Facebook)	255 (57.8)
	Ministry of Health, Bhutan (Facebook)	253 (57.4)
	Information shared in social media groups	45 (10.2)
	Friends, colleagues, or relatives	62 (14.1)
	WHO website	141 (32.0)
P9. During the COVID-19 crisis, where do you usually go?	Hospital	255 (57.8)
	Market	88 (19.9)
	Flu clinic	191 (43.3)
	Hangout with family/friend	31 (7.0)
	Have lunch, or dinner with family/friends	36 (8.2)
	Do not leave the house	156 (35.4)
**Practice score**	N (%)	
Good	369 (83.7%)	
Mean ± SD	24.7 ± 4.8	
Range (min–max)	0–30	
*Note: Scoring system: always = 4, usually = 3, sometimes = 2, rarely = 1, never/don’t know = 0.*		

WHO—World Health Organization; SD—standard deviation.

**Table 4 ijerph-20-02942-t004:** Multiple logistic regression of knowledge, attitude, and practice scores among study participants in Phuentsholing Hospital, Bhutan.

Characteristics		Knowledge		Attitude		Practice	
		AOR (95% CI)	*p* Value	AOR (95% CI)	*p* Value	AOR (95% CI)	*p* Value
Sex							
	Male	Ref		Ref		Ref	
	Female	1.59 (0.995, 2.540)	0.052	1.29 (0.835, 2.008)	0.249	0.74 (0.353, 1.561)	0.432
Age groups (years)							
	18–25	Ref		Ref			
	26–35	1.52 (0.806, 2.865)	0.196	1.42 (0.785, 2.555)	0.248	0.11 (0.026, 0.484)	**0.003**
	36–45	1.06 (0.521, 2.158)	0.871	1.25 (0.637, 2.449)	0.518	0.27 (0.057, 1.306)	0.104
	>45	0.92 (0.419, 2.010)	0.831	0.94 (0.441, 2.003)	0.873	0.12 (0.026, 0.588)	**0.009**
Ethnicity							
	Ngalong	Ref		Ref		Ref	
	Sharchokpa	1.28 (0.677, 2.436)	0.445	0.50 (0.271, 0.934)	**0.029**	0.45 (0.128, 1.601)	0.219
	Lhotshampa	0.88 (0.477, 1.636)	0.693	0.60 (0.327, 1.105)	0.101	0.46 (0.136, 1.547)	0.209
	Khengpa	0.78 (0.286, 2.118)	0.624	0.70 (0.261, 1.857)	0.469	1.36 (0.188, 9.810)	0.762
	Other ethnicities	1.75 (0.336, 9.119)	0.506	0.51 (0.136, 1.913)	0.318	0.08 (0.010, 0.646)	**0.018**
Education							
	Illiterate	Ref		Ref		Ref	
	MNFE	3.82 (1.199, 12.141)	**0.023**	1.87 (0.569, 6.115)	0.304	1.97 (0.495, 7.862)	0.336
	Primary	1.68 (0.486, 5.781)	0.414	1.98 (0.595, 6.570)	0.266	4.43 (0.938, 20.922)	0.06
	Secondary	3.50 (1.425, 8.619)	**0.006**	3.53 (1.454, 8.555)	**0.005**	11.50 (3.439, 38.486)	**<0.001**
	Higher education	9.23 (3.438, 24.797)	**<0.001**	2.97 (1.154, 7.660)	**0.024**	12.31 (2.952, 51.318)	**0.001**
Occupation							
	Civil servants	Ref		Ref		Ref	
	Private/business	0.62 (0.344, 1.124)	0.116	0.96 (0.566, 1.627)	0.879	8.81 (1.615, 41.455)	**0.011**
	Homemaker	0.49 (0.219, 1.091)	0.08	1.04 (0.488, 2.214)	0.92	1.07 (0.298, 3.823)	0.919
	Student	0.61 (0.250, 1.513)	0.29	2.48 (1.010, 6.097)	**0.048**	0.17 (0.031, 0.942)	**0.042**
	Others *	0.41 (0.162, 1.011)	0.053	1.24 (0.517, 2.977)	0.63	0.27 (0.071, 1.011)	0.052
	Farmer	0.32 (0.124, 0.834)	**0.02**	0.34 (0.132, 0.869)	**0.024**	0.32 (0.093, 1.073)	0.065

MNFE—Monastic and non-formal education; Others—monks and armed forces; AOR—adjusted odds ratio; CI—confidence interval; Ref—reference group. * monks/nun, armed force/uniformed personnel, farmer; **bold**—significant at 5% level.

**Table 5 ijerph-20-02942-t005:** Correlation between knowledge, attitude, and practice of COVID-19 of study participants of Phuentsholing Hospital, Bhutan.

Domain	N (%)	Correlation	
Knowledge category		Knowledge	Attitude
Poor knowledge < 60%	197 (44.7)		
Good knowledge ≥ 60%	244 (55.3)		
Attitude Category			
Negative attitude < 60%	217 (49.2)		
Good attitude ≥ 60%	224 (50.8)	**r = 0.228** ***p* value < 0.001**	
Practice category			
Poor practice < 60%	72 (16.3)		
Positive practice ≥ 60%	369 (83.7)	**r = 0.220** ***p* value < 0.001**	**r = 0.338** ***p* value < 0.001**

Note: N = number of participants; r = correlation coefficient; **bold**—significant at 5% level.

## Data Availability

The data supporting the results can be obtained upon request from the corresponding author.

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
