# Peer review of "Knowledge, Attitude, and Practice towards COVID-19 among Patients Attending Phuentsholing Hospital, Bhutan: A Cross-Sectional Study"

_ijerph, 2023, doi:10.3390/ijerph20042942_

Round 1

Reviewer 1 Report

General comments:

This paper study the knowledge, attitude, and practice towards COVID-19 among patients attending Phuentsholing Hospital and identify its associates. The research object of the article is more suitable, the charts are beautifully made, references are appropriate, but the elaboration of scientific issues, the innovation still needs to be improved.

1. Abstract:

The abstract has too much content and the results section needs to be streamlined.

2. Background:

The background section is not well constructed. Normally, the introduction should contain three paragraphs, the first paragraph should illustrate the background of this study. The second paragraph should contain literature review, then point out the shortcomings of existing research, then show your innovation of your study. The third paragraph should include study objectives. However, this introduction  is only to state the history and successful experience of COVID-19 in Bhutan, has no literature review, and you did not point out the shortcomings of existing research, and study objective is not clear.

3. Background:

It can be seen from the literature that a lot of relevant studies have been carried out. This paper only changed the region for research, so it is necessary to clearly explain the representativeness and necessity of your choice of Bhutan for research.

4. Literature citation:

References in a text should use brackets, not small brackets.

5. Discussion:

To the best of my knowledge, the purpose of this paper is to better strengthen the epidemic prevention and control work, so the countermeasures and suggestions are the top priority of this paper. However, the existing documents do not cover this, so it is suggested to supplement the relevant countermeasures and suggestions separately.

Besides, some contributions about this paper should be added to Discussion section.

Reviewer 2 Report

This study covers the KAP towards COVID -19 among patients in Bhutan. 

Although this is a cross-sectional study from a single medical center, I believe that the results are very relevant and important for decision-makers in preparation for the next pandemic anywhere around the world. 

I do have a few questions: 

1.      Is there one KAP-validated questionnaire? Why have you used the knowledge part from Olum grading and the attitude part from Goni (ref 28?)  please further elaborate in Methods- why these questionnaires came from these specific studies. Just elaborate on what was validated in the field before and why the specific tools for KAP assessment were chosen. 

2.     Please discuss the discrepancy between medium knowledge and attitude and the high vaccination rates. 

Significant negative covariates of good practices were those in 26-35 and >45 years 291 and students. Older people engaging in poor practice have been reported in another study 292 (47). While other studies reported good practices in >40 years (13,27,40,48).” Rows 291-293- I find these sentences a bit confusing – please rewrite so the reader will better understand in which age group practices were not good

Reviewer 3 Report

Thank you for letting me review this interesting manuscript.

In the introduction, attention is not focused on solution knowledge, attitude, and practices (KAP). The motivation of this submission is weak. The introduction section should be organized better, so the reader could immediately assess the results and the motivation of the paper. 

I would propose to the authors to write a much clearer paper where the reader could better understand what they are trying to achieve with a new approach and how this

contributes to the state of art in Science.

What is the research question or main research objective that you address in the paper? Which research method do you apply? Provide a research method that shaped your actions in the whole research.

Data collection instruments:

The authors describe: "The study tool comprised an interviewer-administered questionnaire that had five parts". However, the fifth step is not clear.

I'm concerned about sample detail. I suggest authors broaden the content to make it easier for readers to understand.

I recommend reviewing for exemple: "≥18 years". 

In my perception this section "Ethics approval and consent to participate" should not be part of the text of the article, as it is supplementary information.

I recommend using percentage for Figure 2. Distribution of sources of COVID-19 information.

In my view, some countries in the world had problems with fake news, as a lot of unofficial information was presented to the population. In this way, I recommend that authors discuss the extent to which this information on the internet is reliable?

It is only worth mentioning "The COVID-19 Infodemic on Twitter: A Space and Time Topic Analysis of the Brazilian Immunization Program and Public Trust"

The authors presented the following arguments: "As a result, private and business people were affected most by this pandemic" Describe the argument better.

"Overall positive attitude in this study was 51.0%, which is much lower (86.6%)" . I recommend that authors do not present data in section 4.

I recommend that authors answer the objective of the work in the conclusion with accurate information.

In addition, it would be very important for health professionals to understand a new section to describe what are the practical and theoretical implications of the work of this post-pandamic content.

Finally, I appreciate the opportunity to review this submission. While the topic is interesting and worth performing, more alaboration on theory devolpment and motivation is need. I hope the comments are helpful and wish the autors good luck for their paper publication

Round 2

Reviewer 1 Report

1. For the introduction, indent the first line of each paragraph with 2 strings.

2. The format of the references is not written according to the requirements of the journal.

Author Response

  1. For the introduction, indent the first line of each paragraph with 2 strings.

Response 1: Thanks for the comment. We have made the recommended changes in the revised manuscript.

  1. The format of the references is not written according to the requirements of the journal.

Response 2: We have formatted the references as per the journal requirements. Thank you for this comment.
